# Clinical Dilemmas in the Treatment of Elderly Patients Suffering from Hodgkin Lymphoma: A Review

**DOI:** 10.3390/biomedicines10112917

**Published:** 2022-11-14

**Authors:** Vibor Milunović, Ida Hude, Goran Rinčić, Davor Galušić, Aron Grubešić, Marko Martinović, Nika Popović, Sunčana Divošević, Klara Brčić, Marin Međugorac, Luka Kužat, Dejan Strahija, Stefan Mrđenović, Inga Mandac Smoljanović, Delfa Radić-Krišto, Slavko Gašparov, Igor Aurer, Slobodanka Ostojić Kolonić

**Affiliations:** 1Division of Hematology, Clinical Hospital Merkur, 10000 Zagreb, Croatia; 2Division of Hematology, Clinical Hospital Centre Zagreb, 10000 Zagreb, Croatia; 3Division of Hematology, Clinical Hospital Centre Sestre Milosrdnice, 10000 Zagreb, Croatia; 4Division of Hematology, Clinical Hospital Center Split, 21000 Split, Croatia; 5School of Medicine, University of Split, 21000 Split, Croatia; 6Division of Hematology, Clinical Hospital Center Rijeka, 51000 Rijeka, Croatia; 7School of Medicine, University of Rijeka, 51000 Rijeka, Croatia; 8Division of Hematology and Oncology, General Hospital, 42000 Varaždin, Croatia; 9Policlinic Medikol, 10000 Zagreb, Croatia; 10Division of Hematology, Clinical Hospital Sveti Duh, 10000 Zagreb, Croatia; 11Division of Oncology and Hematology, General Hospital Čakovec, 40000 Čakovec, Croatia; 12Division of Hematology, Clinical Hospital Center, 31000 Osijek, Croatia; 13School of Medicine, University of Osijek, 31000 Osijek, Croatia; 14Clinical Department of Pathology and Cytology, Clinical Hospital Merkur, 10000 Zagreb, Croatia; 15School of Medicine, University of Zagreb, 10000 Zagreb, Croatia

**Keywords:** Hodgkin lymphoma, frail elderly, geriatric assessment, ABVD protocol, brentuximab vedotin, immune checkpoint inhibitors

## Abstract

Elderly patients make up a significant number of cases of newly diagnosed Hodgkin lymphoma. However, unlike in young patients, the outcomes of elderly patients are poor, and they are under-represented in phase III trials. Prior to treatment initiation, geriatric assessment should ideally be performed to address the patient’s fitness and decide whether to pursue a curative or palliative approach. The ABVD regimen is poorly tolerated in unfit patients, with high treatment-related mortality. Alternative chemotherapy approaches have been explored, with mixed results obtained concerning their feasibility and toxicity in phase II trials. The introduction of brentuximab vedotin-based regimens led to a paradigm shift in first- and further-line treatment of elderly Hodgkin lymphoma patients, providing adequate disease control within a broader patient population. As far as checkpoint inhibitors are concerned, we are only just beginning to understand the role in the treatment of this population. In relapsed/refractory settings there are few options, ranging from autologous stem cell transplantation in selected patients to pembrolizumab, but unfortunately, palliative care is the most common modality. Importantly, published studies are frequently burdened with numerous biases (such as low numbers of patients, selection bias and lack of geriatric assessment), leading to low level of evidence. Furthermore, there are few ongoing studies on this topic. Thus, elderly Hodgkin lymphoma patients are hard to treat and represent an unmet need in hematologic oncology. In conclusion, treatment needs to be personalized and tailored on a case-by-case basis. In this article, we outline treatment options for elderly Hodgkin lymphoma patients.

## 1. Introduction

The 2016 World Health Organization classification divides classical Hodgkin lymphoma (HL) into four groups based on the immunohistochemistry features and biology [1]. In the US, the age-adjusted incidence of HL is 2.6. per 100,000 people [2]. There are two peaks of newly diagnosed HL according to age—i.e., 30.8% of cases occur in the age group between 20 and 34 years old and 19.5% of cases occur in patients older than 65 years. The five-year relative survival rate is 89.1%, with the median age of death being 70 [2]. While younger patients have a low mortality rate, the majority of deaths occur in elderly patients, with the death rate among patients older than 65 exceeding 35%. Similar findings by the German Hodgkin’s Study Group (GHSG), which included 372 elderly patients from multiple clinical trials, showed a 5-year overall (OS) of 65% among the elderly (HR = 3.9) [3]. Elderly patients with HL are therefore a vulnerable group, warranting an improvement of the current treatment algorithm.

## 2. Geriatric Approach to Aggressive Lymphomas

When making a decision on treatment goals and choosing therapy for elderly patients with cancer, it is advisable to perform a comprehensive geriatric assessment (cGA) through a multidimensional process, such as per the American Society of Clinical Oncology (ASCO) guidelines [4]. Geriatric assessment (GA) tools frequently used in aggressive lymphomas are shown in Table 1: “The Geriatric Approach to Elderly Patients with Malignant Lymphoma”.

As seen in the references above, the majority of data come from patients with more frequent aggressive lymphomas—i.e., diffuse large B-cell lymphoma (DLBCL), commonly treated with the CHOP-R regimen [27]. Currently, we are witnessing a positive trend in successful treatment of older patients with aggressive lymphoma, as shown by a meta-analysis conducted by Tavares et al. [28]. In this particular analysis, however, only 3 out of 38 studies involving very elderly patients (age ≥ 80) with DLBCL focused on cGA. However, the trend is changing, and more studies have focused on this approach in DLBCL, since it is primarily lymphoma of the elderly. Eyre et al. retrospectively examined infection-related mortality in 690 elderly patients, with curative intent being 7.2% at 1 year. In a multivariate analysis, alongside high International Prognostic Index values and albumin levels, a CIRS-G score≥ 6 was the strongest variable to predict this outcome [7]. The G8 scale was associated with poor outcomes in 388 elderly patients (HR for OS = 0.871) [13]. Concerning the loss of IADL, in a retrospective study on 142 patients (age ≥ 80) treated with miniCHOP-R, this factor was the most predictive for death event (HR = 12.39) [18]. These and other studies are “proof of concept” that cGA plays an important role in the treatment and decision making in elderly patients with malignant lymphoma, and we believe that, given the paucity of data in HL, these results from appropriately designed up-to-date cGA studies in DLBCL may be translated to HL. The earliest registry study of historical importance carried out in the Netherlands found that comorbidities were associated with the delivery of treatment to elderly patients with HL [29]. In the modern era, a retrospective multicenter study on 88 elderly patients, the majority of whom were treated with the ABVD regimen, found two risk factors for adverse outcome in multivariate analysis: age ≥ 70 (HR for OS = 2.24) and loss of ADL (HR for OS = 2.71) [30]. The largest study conducted thus far was based on SEER data on 1315 patients receiving the full ABVD regimen and analyzed the risk factors for 1-year mortality [2,31]. Increases in Charlson Comorbidity Index values [32] (OR = 1.41) and age (OR = 1.33) were predictive of a worse outcome. Interestingly, disability status was not found to be a significant predictor, possibly due to patient selection bias (i.e., only fit patients offered ABVD regimen). Orellana-Noia et al. carried out research across 10 medical centers in the US (*n* = 244) and showed that superior outcomes were obtained after treatment with the conventional regimens compared to alternative therapies [11]. In a multivariate analysis, only loss of ADL was found to be predictive of poorer OS (HR = 2.13). Treatment was feasible, with treatment-related mortality (TRM) being 3.3%. However, toxicities did impact the total number of delivered cycles and reduced CR rates accordingly, resulting in lower PFS for patients experiencing them. This study suffers from several limitations. Due to inherent selection bias and retrospective analysis, it does not reflect “real-world” patients. Furthermore, only one patient underwent cGA before treatment initiation, with other functionality and geriatric assessments conducted in a retrospective manner. The latter matter should be assessed prospectively to draw firmer conclusions in HL patients; results from an observational study examining the relationship between cGA, type of treatment used and the long-term outcomes currently underway are eagerly awaited [33]. The cGA of an elderly HL patient is a valuable first step in treatment decision making, recommended by experts in this field [34,35,36]. Such assessment is shown to be more effective than clinical judgement in stratifying lymphoma patients according to fitness status and should be used whenever possible. However, given its laborious nature, it is often not feasible in every day clinical practice. On the other hand, indexes such as ADL [14,15,16], IADL [17] or G8 [12] (for other examples see Table 1) are widely available and manageable within couple of minutes.

## 3. Does ABVD Regimen Treatment Have a Role in Elderly Patients with HL?

### 3.1. Early and Intermediate Disease

GHSG HD 10 and HD 11 phase III randomized clinical trials (RCTs) have established the historical standard of care for younger patients [37,38]. A sub analysis of HD10 and HD11 included 177 elderly patients, with the majority of patients receiving four cycles of ABVD (93%) [39]. The complete remission (CR) rate was high (89%), and 5-year OS and PFS rates were 81.2% and 74.8%, respectively. An additional analysis of HL-specific outcomes was undertaken, showing higher rates of OS and PFS and suggesting the existence of various competing factors influencing survival. One of the possible factors was the high rate of grade III and IV adverse events (AEs), which was 68%. Infections and respiratory disorders were more prevalent in this population in comparison to younger patients, and treatment-related mortality was 5%. Moreover, toxicities probably influenced relative dose intensity (RDI) of chemotherapy (with 41% of patients having an RDI < 80%), with frequent delays in therapy seen. Concerning other treatment modalities, IFRT was found to be feasible, with mild toxicity. Patients who received two cycles of ABVD experienced fewer AEs with a greater RDI; however, the final outcomes were similar (for details, see the appendix of the original study) [39]. On the other hand, feasibility of the BEACOPP baseline regimen was low in terms of the high rate of AEs, more protocol deviations and TRM favoring ABVD (see the appendix of the original study) [39]. The HD13 trial examined the role of omission of dacarbazine and/or bleomycin for early-stage HL and found that it compromises efficacy and should not be generally recommended [40]. Subsequent analysis from the same group included 287 older patients from HD10 and HD13 trials and stratified them by regimen received: ABVD (two or four cycles) and AVD (two cycles) [41]. The CR rate was high and similarly independent of the regimen used. The PFS and OS rates in elderly patients, however, seem to be inferior to those reported in younger patients. Considering only the results extrapolated from the HD13 trial (two cycles of ABVD vs. two cycles of AVD), there seems to be no difference in final outcomes (HR for PFS = 1.28; HR for OS = 1.3); however, we must note that the 95% CI are wide, leading to the low level of evidence, and as the authors stated, the analysis was underpowered to draw a noninferiority conclusion. Concerning feasibility, the RDI was like that of younger patients, with the early termination rate being low. The rates of grade III or IV AEs were similar whether receiving ABVD or AVD (40% vs. 39%, respectively), with negligible TRM. The authors concluded that the application of a maximum of two cycles of ABVD is feasible in older patients; however, they stated that the omission of bleomycin is mandatory when more than two cycles are used.

### 3.2. Advanced Disease

Concerning ABVD in advanced disease, randomized data are available from the ECOG E2496 phase III RCT comparing ABVD vs. Stanford V regimen [42]. An additional analysis was undertaken for elderly patients [43]. Since this trial established ABVD as the regimen of choice at the time of publication, we focus on the ABVD arm (*n* = 23). The ORR was 74%. Concerning outcomes, the 5-year failure-free survival (FFS) rate was 53%, and the 5-year OS rate was 64%, showing a statistically significant difference (*p* = 0.002 and *p* < 0.001, respectively) when compared to the rates in younger patients. The grade III or IV AE rate was 92%. AEs of clinical interest were febrile neutropenia and respiratory disorders (as discussed in the section “Should We Be Afraid of Bleomycin Use in Elderly Patients with HL?”), with TRM being 8%. RDI was 73% in this population, showing a poor feasibility. The authors concluded that novel approaches are needed to improve tolerability and efficacy in advanced disease setting. In advanced disease, the role of ABVD is controversial. However, if practicing physicians decide to use ABVD in advanced stage disease, our recommendation is to implement an interim positron emission tomography (iPET) adapted strategy, such as per RATHL trial design [44]. Patients with positive iPET results should not continue with the same treatment protocol, given the anticipated lack of efficacy and unacceptable toxicity, but could rather be offered enrollment in clinical trials, alternative treatment or receive best supportive care. Although there has been no formal analysis in this particular study concerning elderly patients, the retrospective study from Bentur et al. represents a “proof-of-concept” study on iPET [45]. Concerning treatment, most subjects were treated using the ABVD protocol (67%), with most patients achieving complete metabolic remission (CMR) as per Lugano criteria [46]. The iPET positivity was associated with poor outcomes in terms of PFS and OS. In a regression analysis, the only predictor for poor outcomes was positive iPET (HR for progression = 8.5, HR for death = 6.9). The study suggests that iPET is useful in guidance of the treatment goal, i.e., to continue with curative intent or palliative care. This notion is further confirmed by the sub analysis of the GATLA LH-05 trial [47,48]. Treatment with three cycles of ABVD and iPET CMR had excellent outcomes, with median PFS and OS not reached. More importantly, no grade III and IV AEs were noted, representing a possible treatment option for a subset of elderly patients.

The data obtained on ABVD regimen are presented in Table 2: “ABVD Regimen in Elderly Hodgkin Lymphoma.”

Published trials suffer from multiple limitations. The first and the most important one is selection bias, meaning that only a low number of patients were included, not representing the “real world” population of elderly HL. The main reason for this is rigorous inclusion criteria, mainly fitness and adequate organ function. Yet, as shown in Table 1, toxicities are common even in fit subjects, sending an important safety signal. Furthermore, it is evident that extrapolation of the treatment of younger patients is not possible to the elderly, primarily due to AEs occurring in this setting.

## 4. Should We Be Afraid of Bleomycin Use in Elderly Patients with HL?

Bleomycin-induced lung injury (BILI) is a syndrome that presents with a variety of clinical features (ranging from acute respiratory distress syndrome to pneumonitis or pulmonary edema), with diagnosis being based on the exclusion of other possible respiratory disorders [49]. Based on registry data obtained from 835 patients with germ cell tumors, the contributing risk factors for bleomycin toxicity are age greater than 40 years (HR = 2.3) and impaired glomerular filtration rate (HR = 3.3) [50]. In our opinion, these findings readily translate to HL patients.

The actual incidence of BILI in elderly patients with HL is not known, with variable rates reported due to the use of retrospective study designs and reporting biases [11,30,43,51,52]. Concerning prospective trials, only Evens et al. used exact methods for testing pulmonary lung function [43]. In total, 10 patients treated with ABVD developed BILI, with AE commonly being grade I or II, with two death events. The experience from the Mayo Clinic shows the rates of BILI and BILI-related mortality being 18% and 24%, respectively [51]. Patients with BILI had a median 5-year OS rate of 63% vs. 90% in unaffected patients (*p* = 0.001). However, there was no significant HL-related OS difference among patients surviving bleomycin toxicity compared to ones not developing it, suggesting that the survivors of BILI have the same long-term outcome. Due to the small patient sample, multivariate analysis was not performed, but univariate analysis suggests age greater than 40, the use of G-CSF factors and the application of the ABVD regimen as possible risk factors. French data on elderly patients (*n* = 117, with the majority having advanced disease and the mean N of ABVD cycles = 6) reported the rate of occurrence of pulmonary toxicity of grade III or IV [52]. Early pulmonary AEs occurred in 31 patients, with 7 death events, while late pulmonary AEs occurred in 22 patients. In total, the mortality rate related to pulmonary events was 22%. Despite some of these reports suggesting no influence on HL-related outcomes for survivors of acute toxic events, one must keep in mind that long-term consequences can severely impair the survivor’s quality of life.

In conclusion, the data on BILI in older patients are strong enough to question the use of bleomycin in the treatment algorithm. If used, it should be given cautiously and by an experienced clinician, able to recognize and treat BILI promptly if it occurs.

## 5. ABVD and Leading Guidelines

Recently published British guidelines recommend that GA should be undertaken prior to deciding on treatment, using GRADE criteria [36,53]. Concerning fit patients, they suggest the use of the AVD regimen as a treatment of choice, especially in early stage (1B). Furthermore, a maximum of three cycles of ABVD may be used as part of the iPET adapted strategy for advanced disease (2B). It is important to note that cardiac comorbidity must be assessed before initiating a doxorubicin-containing regimen due to the high rate of cardiovascular mortality [34,35,36]. The option recommended by the National Comprehensive Cancer Network (NCCN) guidelines for early, favorable disease is two cycles of A(B)VD followed by irradiation [54]. For advanced disease, the advised course is two cycles of ABVD followed by four cycles of AVD in the case of iPET negativity, as shown by the RATHL study [44]. Furthermore, when using the ABVD regimen, the use of G-CSF is not recommended due to higher incidence of BILI. Concerning the European Society for Medical Oncology guidelines on HL, the authors state that ABVD-based chemotherapy is the therapy of choice, with omission of bleomycin after two cycles (grade III, B–C) [55]. The main emphasis of these guidelines’ recommendation is limiting toxicity in first-line treatment.

The first part of the approach in treating these patients with ABVD is multidisciplinary cGA to identify fit patients who may benefit from ABVD. Concerning early or intermediate disease, we recommend two to four cycles of A(B)VD followed by IFRT. Omission of bleomycin remains at the treating physician’s discretion; however, its use should not exceed two cycles of treatment [41]. In advanced disease, six cycles of ABVD are unfeasible and should not be used in this setting. If a treating physician chooses ABVD, it should follow PET-adapted strategy such as the RATHL approach with modification that iPET positive patients should be referred to clinical trial, if possible, or palliative care [44].

## 6. Alternative Chemotherapy Regimens

In an attempt to improve outcomes and/or avoid the toxicities of ABVD, the scientific community has developed alternative chemotherapy regimens. The GHSG found BEACOPP baseline and BACOPP regimens unacceptably toxic and without any improvement in outcomes of older patients [56,57]. Concerning palliative care, ChlVPP and PEP-C are regimens of choice [58,59]. Several other regimens have intrigued the hematological community. The CHOP21 regimen was tested in a small single-center retrospective study (*n* = 29) followed by IFRT [60]. The stratification of patients was based on stage and risk factors. For the whole group, median OS was not reached, with the 3-year PFS rate being 76% and without significant difference according to stage. Toxicity was high due to the lack of supportive care, with two grade V events. The retrospective design and small N of patients are a mayor limitation of this study. GHSG developed the PVAG regimen and examined it on 59 fit patients [61]. The use of the regimen was found to be feasible, with the RDI being 88%, meeting the primary endpoint. The CR rate was 78%, translating to 3-year PFS and OS rates of 58.4% and 66.1%, respectively. Concerning toxicity, infections were predominant, as well as respiratory tract AEs attributed to gemcitabine. Recently, the LYSA group presented their own results with PVAG for 49 patients [62]. The main difference from the original study is the different population used, i.e., older, frail patients with more comorbidities validated by the CIRS-G tool, as well as the use of modified dosing of PVAG [5]. The CMR rate was 53%, with the median PFS and OS being 21.6 months and 66.5%, respectively. Toxicity was manageable, with febrile neutropenia being common. In a univariate analysis, higher CIRS-G grade in ≥two categories was found to be a significant predictor for inferior OS (HR = 3.63). Unfortunately, the data from the multivariate analysis are not available in the original text. This regimen is certainly valuable in frail patients [34,35,36]. A phase II trial on the VEPEMB regimen stratified 105 older patients by stage and the presence of B symptoms [63]. In favorable disease, the use of VEPEMB was found to be feasible, with most patients completing the treatment plan, while in the advanced stage only 60% of patients received the full regimen. One of the main reasons for this was unacceptable toxicity due to unknown reason and low RDI, leading to statistically significant inferior outcomes in terms of 5-year FFS (34% vs. 79%) and 5-year OS (32% vs. 94%). To assess this issue, RCT was developed, but due to poor accrual it was stopped prematurely.

Extrapolating most of these data to a “real-world” setting is difficult due to aforementioned selection biases. In our opinion, careful stratification of patients, especially according to GA and possible contraindications for certain agents, is necessary in order to guide practicing physicians in choosing the appropriate first-line treatment.

The data on the selected alternative chemotherapy regimens are summarized in Table 3: “Alternative Chemotherapy Regimens with Curative Intent in Elderly Patients with Hodgkin Lymphoma”.

## 7. The Use of Novel Agents in the Treatment of Elderly Patients Suffering from HL

### 7.1. Brentuximab Vedotin

Brentuximab vedotin (BV) has led to a paradigm shift in our approach towards younger patients with HL in multiple clinical scenarios, justifying its inclusion in treatment regimens for elderly patients [64]. Furthermore, most of the research published on brentuximab has included GA to allow stratification by frailty status. Of note, despite being considered more tolerable than standard chemotherapy, its use is not without risks, with peripheral neuropathy of special concern, especially among the elderly.

### 7.2. BV in Frail Patients

BV monotherapy has been addressed in three different trials [65,66,67]. The data obtained show limited activity of BV monotherapy, warranting its use in combination regimens. On the other hand, BV monotherapy represents a possible palliative care option, although PEP-C and ChlVPP have financial and toxicity advantage [58,59]. The use of a combination of BV plus dacarbazine (DTIC) was tested in 22 frail patients [68]. Sixty-two percent of patients achieved CR, median OS was not reached and PFS was 17.9 months for this study arm. A high discontinuation rate was observed due to AEs, mainly peripheral neuropathy (PN). The same study examined the role of adding bendamustine; however, due to unacceptable toxicity and high TRM, this arm was terminated early [67,68]. The HALO trial also examined the use of the latter regimen in 49 frail patients [69]. The 2-year PFS and OS rates in the intention-to-treat population were 54% and 83%. The main difference between the two studies on BV-bendamustine is the different dosing schedules (six cycles of 1.8 mg/kg BV + 90 or 70 mg/m^2^ bendamustine, followed by optional BV monotherapy in the first vs. six cycles of 1.2 mg/kg + 90 mg/m^2^ in the latter study). Concerning toxicity, cytomegalovirus (CMV) reactivation was often, requiring valganciclovir treatment, and four patients died due to CMV infection. Regular measurement of CMV viral load and preemptive valganciclovir treatment are therefore mandatory in the case of BV-bendamustine treatment.

### 7.3. BV in Fit Patients

The ECHELON-1 trial established the BV + AVD regimen as a safe and efficacious frontline option in advanced disease [70]. Evans et al. performed an analysis of its role in fit elderly patients (*n* = 186) [71]. Concerning outcomes, the 2-year PFS for BV-AVD vs. ABVD group were similar, 73.8% vs. 68.9% (HR = 0.84., *p* = n.s.), however, noting this analysis was underpowered to test for superiority. The AE of special concern in the experimental arm was febrile neutropenia, prior to protocol amendment and the inclusion of G-CSF as primary prophylaxis. PN was common in patients treated with a BV-based regimen (any grade 65%), with resolution being reached in most of the patients after treatment cessations. Since BV + AVD has a similar efficacy compared to ABVD in this group of patients, NCNN guidelines do not recommend its use [54]. In our opinion, this regimen should not generally be used in elderly patients, with possible exceptions. GHSG and the Nordic Group developed the new regimen BV-CAP, which was tested on 49 patients (*n* = 49) [72]. Concerning toxicity, febrile neutropenia was prominent, and one patient died due to infectious complications. After a short follow-up, the outcome data assessed at 1 year showed a PFS of 73.9% and an OS of 92.6% [34].

### 7.4. Sequential Therapy

Sequential therapy (BV induction + AVD + BV consolidation) was tested on 48 patients regardless of their level of frailty, with the idea of BV “priming” the lymphoma before introduction of AVD, as well as minimizing overlapping neurotoxicity [73]. Half of the study population completed the whole course of therapy, with the majority completing AVD. A high rate of CR (96%) was reported after AVD, meeting the predefined endpoint, with the 2-year PFS and OS rates being 84% and 93%, respectively, indicating this regimen is highly effective in this setting. Most grade III or IV AEs were infections, and most PNs reported were grade II (27%). Concerning GA, the majority of patients were frail, and loss of IADL was found to be a significant prognostic factor for inferior PFS (HR = 13) and OS (HR = 11.64). In further analysis, the loss of IADL was found to independently predict a lower CR rate, leading to inferior 2-year PFS (25% vs. 94%) and OS (67% vs. 97%) probably due to the low RDI (*p* = 0.01) and AEs (*p* = 0.2) [74]. These results are promising in the elderly HL patient population, and represent a “proof of concept”, confirming the importance of GA.

### 7.5. Brentuximab Vedotin: A New Role in First-Line Treatment?

BV-based immunochemotherapies are highly attractive in this setting, representing treatment options both in fit and frail subjects ineligible for standard chemotherapy. Yet, one of the major pitfalls of these studies is the small number of patients included. However, high rates of discontinuation are observed due to toxicities, mainly PN, compromising its efficacy and quality of life of these patients, indicating that a physician experienced with antibody-conjugates should carry out this treatment in order to minimize this AE. As stated above, other regimens have their particularities. BV + AVD is an intensive protocol, with febrile neutropenia being of concern and warranting supportive care in terms of primary G-CSF prophylaxis [71]. There is a certain ambiguity about BV-B regimen due to its AEs, yet the HALO trial has shown feasibility with CMV prophylaxis [69]. The most controversial is certainly BV monotherapy, not offering adequate disease control except in terms of PFS, suggesting palliative purpose but also bringing the cost-effective aspects’ of such an approach into question [65,66,67]. Certainly, sequential therapy is the most popular among physicians, but the true candidates must carefully be selected due to loss of IADL being a major adverse factor for poor outcomes [73]. The small number of patients and inherent biases of published studies prevent us from drawing very firm conclusions. Further and more extensive research on different BV combination regimens is needed in order to gain broader regulatory agency approval in this setting. In line with this, British guidelines have not yet adopted BV-based therapies in treatment algorithms, while the NCCN suggests the use of BV-DTIC and sequential therapy [36,54].

The data on brentuximab vedotin-based combinations are presented in Table 4: “The Role of Brentuximab Vedotin in Elderly Patients Suffering from Hodgkin Lymphoma”.

## 8. Checkpoint Inhibitors: A Chemo-Free Approach?

Nivolumab and pembrolizumab, the anti-PD1 ligands, have profoundly influenced the treatment paradigm in r–r HL patients, arousing interest for their use in first-line setting, especially in the treatment of patient’s ineligible for chemotherapy such as the elderly. It is important to note that only nivolumab has been tested in frail subjects. In a small phase II trial (*n* = 22), nivolumab in combination with BV was given for up to 16 cycles [75]. The CMR rate was 73%, with median OS and PFS not being reached at time of data cutoff. Immune-related AEs were rare. The rate of discontinuation was minimal. However, the ACCRU trial did not replicate these results with eight cycles of the same regimen given, not meeting the primary endpoint (ORR ≥ 73%) and terminating accrual after an interim analysis [50,76]. The ORR among 46 evaluable patients was 64%, with the CMR rate being 52%. The median PFS was 18.3 months and the median OS not reached. The discontinuation rate was 13%, mostly due to immune events and PN. This chemo-free combination treatment is certainly interesting. Nivolumab was also tested as monotherapy in elderly unfit patients in a LYSA NIVINIHO study on 61 frail patients [77]. The induction phase consisted of six applications with additional consolidation (18 cycles) in case of CMR. In patients with PR or SD vinblastine was added to the consolidation. ORR after induction was 51.9% with nine CMRs. Median PFS was 9.8 months, while the median OS was not reached (2-year OS rate = 76.7%). AEs related to nivolumab were recorded in 36 patients, mainly immune-related events with a high rate of discontinuation. Unfortunately, these data show limited efficacy of nivolumab monotherapy. Yet, all this research is intriguing and should be the basis for future trials on a chemo-free approach in elderly HL patients.

## 9. Relapsed/Refractory Setting in Elderly Suffering from Hodgkin Lymphoma

### 9.1. Is There a Role for High-Dose Therapy and Autologous Stem Cell Transplant?

As shown above, a significant number of elderly patients with HL will be refractory to experience relapse after first-line therapy. This unfortunate scenario puts the practicing physician into a dilemma whether autologous stem cell therapy (ASCT) may play a role in this setting. The first evidence comes from a small study of small single-center experiment by Puig et al. [78]. The authors identified 15 elderly patients receiving the procedure compared to 137 younger patients. The most commonly used salvage therapy was GDP (gemcitabine, dexamethasone and cisplatin) in older patients, with ORR being 93%. It is important to note that PET-CT imaging was not used as response criteria. The consolidation regimen consisted of etoposide and melphalan. Concerning engraftment, there was no difference in neutrophil recovery, while platelet recovery was statistically significantly prolonged in the older group. Toxicity was favorable, with the most common SAE being neutropenic fever (no difference among the groups) without TRM. This approach resulted in adequate disease control with 3-year PFS and OS rates being 73% and 88%, without statistical significance when compared to younger patients. Yet, this study suffers from major pitfalls, i.e., selection bias, because it is unclear how the elderly patients were selected for ASCT. We must assume that they were fit and without significant comorbidities, making these findings unreproducible to a general population of elderly HL. A French group retrospectively analyzed 128 fit patients (autographed by BEAM regimen (BCNU, cytarabine, etoposide and melphalan) [79]. The most common AE grade III or IV was mucositis (N = 13), while other toxicities of interest were sepsis (N = 3) and pneumonitis (N = 3). The ASCT provided favorable disease control, with the 5-year PFS and OS rate being 54% and 67%, respectively. Interestingly, in multivariate analysis no relevant factor, including disease status, was found, which may contribute to the missing data that are the main limitation of this study. Other retrospective data come from the GELTAMO group, including 121 patients older than 50 years of age [80]. Patients were stratified by age, with 42 of them being of age ≥ 60. Additional analysis was performed for HCT-CI and the Charlson Comorbidity Index (CCI) [5,32]. There was no difference between the groups concerning engraftment, while toxicities of grades III or IV were correlated with higher HCT-CI score, being predominantly pulmonary (21%). Four grade V events occurred. Long-term outcomes were favorable, with the 10-year PFS and OS rates being 51% and 57% without difference among the age groups. In multivariate analysis excluding pre-ASCT disease status, greater scores of HCT-CI and CCI were the only factors associated with poor outcomes, indicating that comorbidities, not age, should guide physicians in decision making about the ASCT procedure in itself. Unfortunately, no study on CGA in this setting is available. Furthermore, currently we are witnesses of increased use of ASCT in the elderly, especially in multiple myeloma and NHLs, which is certainly contributing to better supportive care during the procedure. We are of the opinion that ASCT should be offered to fit r-r HL elderly patients younger than 70 years in second CMR after CGA. CGA should be multidisciplinary, including the gerontologist, geriatric nurse, treating hematological and stem cell transplant hematologist to weigh pros and cons for the ASCT procedure in a particular patient to avoid TRM and define the best option of how to treat a particular patient. The further issue remains in defining the best salvage regimen with high efficacy and low toxicity without compromising the patient status. In the literature review, no such regimen has been defined in this group of patients.

### 9.2. The Role of Novel Therapeutical Options: Are We There Yet?

#### 9.2.1. Brentuximab Vedotin

There are several positions where BV can be incorporated in this setting. The most attractive one is certainly BV consolidation after ASCT in high-risk patients. Although the exact number of elderly patients in the AETHERA study is not reported, elderly patients were involved in this clinical trial [81]. The trial defined high-risk patients as having an early relapse, extranodal disease or primary refractory disease. In the ITT population, median PFS was 42.9 months vs. 24.1 months in the placebo group (HR = 0, 55). However, in a subgroup analysis, patients older than 45 years of age did not have the same benefit as the younger patients, although the analysis itself is underpowered. Due to the fact that elderly patients are more prone to PNs, the careful monitoring of this AE is recommended in order to maintain adequate QoL. The second option lies in the palliative setting using BV monotherapy, if not previously given as first-line therapy as outlined above, although due to toxicities, high discontinuation rate and cost effectiveness (off-label use), one may be prone to give the best supportive care with inexpensive palliative treatment such as ChlVPP or PEP-C [58,59,65,66,67]. One of the options may be a retreatment strategy in elderly patients with BV [82]. In a small study (N = 21), this strategy has been proven as feasible, yet the PN remains of concern due to the cumulative effect of BV, and we do not recommend this approach in elderly patients. In conclusion, BV as post-ASCT consolidation in high-risk HL patients may be feasible, while the other options with this compound seem to be unsatisfactory.

#### 9.2.2. Checkpoint Inhibitors

##### Nivolumab

Unfortunately, it is not possible to comment on the activity of nivolumab in elderly patients with r-r HL. A first study on PD-1 inhibition with nivolumab in a relapsed setting following ASCT and BV did not include elderly patients [83]. In the largest trial CHECKMATE 205 (N = 243), only 15 elderly patients, most of them being in cohort with previous ASCT and BV treatment, were included [84]. Due to the fact that their outcomes were not analyzed, this makes us unable to withdraw any meaningful conclusion. Furthermore, no real-life experience showing its efficacy is published, to our knowledge. However, due to the designation it is commonly given in this setting, unfortunately only data on safety are published [85]. The US registry analyzed 126 cases of AEs related to checkpoint inhibitors, mainly nivolumab, stratified by age, including 53 elderly HL patients. In this group, the rate of hospitalizations (79%) and infections (40%) was significantly higher than previously reported. These data provide an important safety signal that we are only at the beginning of understanding how to use this drug without compromising safety.

##### Pembrolizumab

The KEYNOTE-204 trial included 300 patients relapsing after or being ineligible for ASCT to either receive BV or pembrolizumab until 35 cycles or progression [86]. For the whole group, it was a positive trial showing a PFS benefit in the pembrolizumab group (median PFS 13.2 vs. 8.2 months, HR = 0.65). However, it included 49 patients 65 years of age, and their outcomes were analyzed by the authors. In preplanned analysis, pembrolizumab was more effective in this age group, with median PFS being 8.2 months vs. 5.5 months in the BV group (HR = 0.64). In further analysis, the greatest benefit was observed in subjects ≥ 75 years of age (N = 11, median PFS 8.3 months vs. 3.5 months, HR = 0.25), while the effect was less pronounced in patients in the younger elderly subjects’ group (≥ 65 to < 75, N = 26, median PFS 8 vs. 5.3, HR = 0.79). Yet, it is important to note that this trial did not include frail subjects, with only one patient having ECOG 2. Furthermore, the number of included elderly patients is too small to draw definitive conclusions. Concerning safety in the whole study population, grade III and IV AE events were pneumonitis in the pembrolizumab group (N = 6), while BV was tolerated adequately with only five cases of PN grade III or IV. One grade V event, pneumonia, occurred related to pembrolizumab. Despite limitations stated above, pembrolizumab may be the treatment of choice in elderly fit patients not eligible for ASCT. 

### 9.3. The Role of Novel Therapeutical Options: Are We There Yet? No…

As shown above, elderly patients are not included; all total N is small in clinical trials in the r-r HL setting. Furthermore, frail patients are excluded from these trials despite being a predominant population. In the analysis of outcomes, only the KEYNOTE-204 trial analyzed the effect of therapy in older age groups, while the data for nivolumab are unavailable due to not including this population. The main question on targeted therapy in r-r elderly HL patients remains unanswered due to the fact that extrapolation of these trials in a “real-world” setting is not possible and may lead to unwanted consequences.

## 10. Other Approaches

Most experience regarding r-r Hodgkin lymphoma in the elderly (N = 105) comes from retrospective historical (1993–2007) GHSG analysis [87]. The patients were stratified according to disease status at the beginning of second-line treatment and Josting’s prognostic score (presence of anemia, early relapse and advanced stage) [88]. It is important to note that multiple different modalities were used with curative or palliative approaches (N = 31). For the whole group, 3-year OS rate was 31%, with the majority dying from HL. Concerning Josting’s prognostic score, in this analysis it was dichotomized with the high-risk group having a median OS of only 9 months compared to 45 months in the low-risk group. Multiple treatment modalities were used: intensive chemotherapy, polychemotherapy with salvage radiotherapy and palliative care. Intensive chemotherapy was used in advance with intent to transplant, however, only five patients proceeded to ASCT, with median OS for this subgroup being 10 months. Palliative care included monotherapy, radiotherapy and best supportive care, with median OS being 7 months. Polychemotherapy accompanied with radiotherapy yielded the best results in terms of disease control, with median OS being 41 months. This may be attributed to the fact that this group of patients had low-risk features according to Josting. However, though these data are valuable, we must interpret them with caution due to retrospective design, the historical era and the lack of appropriate supportive care. In conclusion, a subset of low-risk patients, unfit for ASCT, may be salvaged with polychemotherapy or radiotherapy.

## 11. Conclusions: Is There Evidence-Based Medicine in the Treatment of Elderly Patients Suffering from Hodgkin Lymphoma?

The proposed regimen for the first-line treatment of elderly HL patients is presented in Figure 1.

Validated GA tools should be used to determine the patient’s frailty status prior to deciding on treatment goals [4,5,12,14,17]. It is necessary to choose the regimen of treatment on a case-by-case basis and provide the patients with adequate supportive care, including corticosteroid pre-phase and GCSF prophylaxis when appropriate. ChlVPP or PEP-C are the treatments of choice in a palliative setting, with BV monotherapy used in selected patients [58,59,65,66,67]. Concerning early disease, we recommend limiting the use of bleomycin in the ABVD regimen to two cycles or omitting it if contraindicated. [36,41,54]. Other possible regimens that can be used in this setting are listed in Figure 1 [60,61,62,63]. Concerning advanced disease, BV-based immunochemotherapies are the treatment of choice if available, though alternative chemotherapy regimens can be applied [60,61,62,68,70,72,73]. The BV-bendamustine regimen must be delivered with great caution along with adequate supportive care [73]. The use of the interim-PET-CT-based approach is attractive due to the possible omission of bleomycin in later cycles after initiating treatment with ABVD, and other modifications made in the case of positive iPET results (palliation or enrollment into a clinical trial) [48].

Concerning the r-r refractory setting, the most data involve fit patients. High-dose chemotherapy followed by ASCT plays the treatment option in carefully selected patients by multidisciplinary teams. Concerning targeted agents, BV consolidation may be a choice after ASCT in high-risk patients with careful monitoring for PN AEs. Checkpoint inhibitors are attractive, however, there are no data for nivolumab in this setting, while pembrolizumab may represent a treatment of choice in the fit elderly population not ineligible for ASCT. Polychemotherapy or radiotherapy provide certain disease control in low-risk patients. Frail patients with unfavorable prognostic factors in this setting should be referred to palliative care.

However, despite constituting a high rate of HL cases, the elderly population is under-represented in clinical trials [2]. There are multiple biases in this area (study design, selection, reporting and publication biases), leading a to low level of evidence. Future trials must address these issues to provide appropriate data and evidence. Unfortunately, at present there are only a few ongoing phase II trials in this setting, with the prospect of evidence-based medicine in elderly HL patients remaining unfulfilled [89].

## Figures and Tables

**Figure 1 biomedicines-10-02917-f001:**
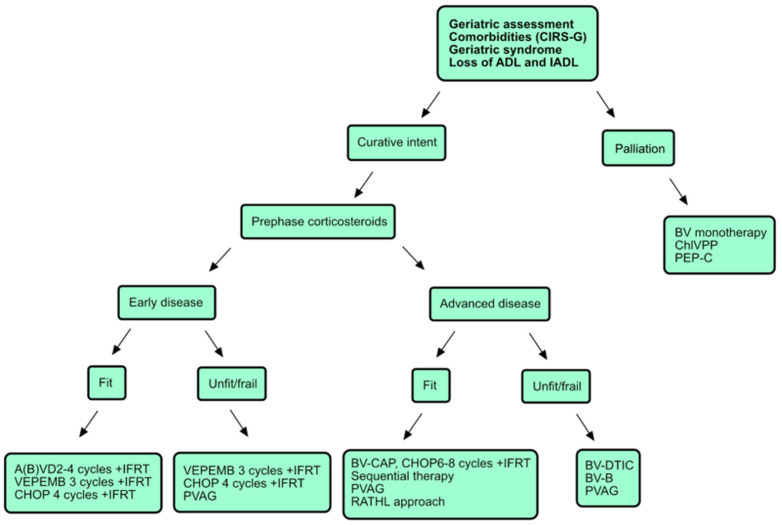
Proposed Treatment Algorithm for the First-Line Treatment of Elderly Patients with Hodgkin Lymphoma. Abbreviations: CIRS-G—Cumulative Illness Rating Scale-G; ADL—activities of daily living; IADL—instrumental activities of daily living; A(B)VD—adriamycin, bleomycin, vinblastine and dacarbazine; IFRT—involved-field radiation therapy; VEPEMB—vinblastine, cyclophosphamide, procarbazine, bleomycin, prednisone, mitoxantrone and etoposide; CHOP—cyclophosphamide, doxorubicin, vincristine and prednisolone; PVAG—prednisone, vinblastine, doxorubicin and gemcitabine; BV—brentuximab vedotin; ChlVPP—chlorambucil, vinblastine, procarbazine and prednisolone; PEP-C—prednisolone, etoposide, procarbazine and cyclophosphamide; BV—brentuximab vedotin; BV-CAP—brentuximab vedotin, cyclophosphamide, doxorubicin and prednisolone; RATHL—Response-Adjusted Therapy for Advanced Hodgkin Lymphoma; BV-DTIC—brentuximab vedotin-dacarabazine; BV-B—brentuximab vedotin-bendamustine.

**Table 1 biomedicines-10-02917-t001:** The Geriatric Approach to Elderly Patients with Malignant Lymphoma.

Domain	Explanation	Tools Used in Aggressive Lymphomas and Proposed by ASCO
Non-cancer life expectancy	Comorbidities predicting the non-cancer-related mortality	CIRS-G [5,6,7] * HCT-CI [8,9]
Geriatric syndrome	The aggregate of “symptoms and signs associated with any morbid process, and constituting together the picture of the disease” [10]	GA [11] ** and G8 [12,13] ***
Function	Loss of activities of daily life (ADL) and instrumental activities of daily life (IADL)	Katz index for ADL [14,15,16] Schonberg index for IADL [17] ****; other indexes [16,18,19,20]
Falls	Number of falls in the last 6 months [4]	Single-item question
Cognition	Cognitive impairment is associated with worse outcomes [4]	Various indexes [21,22,23] *****
Depression	Depression in cancer patients is related to multiple variables, ranging from treatment feasibility and mortality to functional decline [4] ******	GDS [24]
Nutrition	Unintentional weight loss and BMI are associated with mortality [4]	SMM, CT and PET-CT [25] *******
Toxicity	Risk factors for treatment-related toxicity	GA [11], G8 [12], GRI [22], nutrition [25] and CARG toxicity tool [26]

Abbreviations: ASCO—American Society for Clinical Oncology; CIRS-G—Cumulative Illness Rating Scale-Geriatric; HCT-CI—Hematopoietic Cell Transplantation-Specific Comorbidity Index; GA—geriatric assessment; G8: the G8 (Geriatric 8) Health Status Screening Tool; ADL—activities of daily Life; IADL— instrumental activities of daily life; GDS— Geriatric Depression Scale; SMM—Skeletal Muscle Mass; CT—Computed Tomography; PET-CT—Positron Emission Tomography-Computed Tomography; GRI—Global Risk Indicator; CARG: Cancer and Aging Research Group. Footnotes: * value of CIRS-G score as a prognostic factor in multivariate analyses varies; ** not predictive in first-line treatment of HL; see text for explanation and Orellana-Noia [11]; *** simplified tool used to assess GS; **** index for IADL proposed by the ASCO guidelines that has not been studied in aggressive lymphomas; ***** only two indexes have been validated in a population with cancer; ****** based on reviewed research, where depression was studied in various aggressive lymphomas, but there is no study focusing on elderly patients; ******* mixed results found regarding its prognostic value due to the modality used—in DLBCL studies conducted in patients receiving the CHOP-R regimen, it had a clear value for use in prognosis (study type: meta-analysis).

**Table 2 biomedicines-10-02917-t002:** ABVD Regimen in Elderly Hodgkin Lymphoma.

Study	*n*	Disease Stage	Regimen	CR	Outcome Rate *	*p* **	AE Rate N (Rate) ***	CVD-Related Death N (Rate)	Respiratory AE N (Rate) ***	Infection AE N (Rate) ***	TRM Due to AEs
Boll et al. [39]	117	early	ABVD 4x	89%	5-year PFS = 74.8% 5-year OS = 81.2%	*p* < 0.001	79 (68%)	8 (7%)	6 (5%)	11(10%)	5%
Boll et al. [41] ****	287	early	AVD 2x (HD13, N = 82)	98%	5-year PFS = 79% 5-year OS = 91%	*p* = NR	31(40%)	NR	NR	2(3%)	1%
			ABVD 2x (HD13, N = 67)	99%	5-year PFS = 78% 5-year OS = 86%		26 (42%)	3 (4%)	1 (2%)	4 (6%)	NR
			ABVD 2x (HD10, N = 70)	96%	5-year PFS = 79% 5-year OS = 84%		24 (37%)	NR	1(2%)	5 (8%)	3%
			ABVD 4x (HD10, N = 68)	88%	5-year PFS = 79% 5-year OS = 87%		45(65%)	5 (7%)	7 (10%)	(8) 12%	6%
Evens et al. [43] *****	24	advanced	ABVD x6	65%	5-year FFS = 53% 5-year OS = 64%	*p* = 0.002 *p* < 0.0001	22 (92%)	NR	Grade 3 = 5 (20%) Grade 4 = 7 (27%)	Grade 3 = 10 (4%) Grade 4 = 10 (4%)	9%

Abbreviations: N—number, CR—complete remission, AE—adverse event, TRM—treatment-related mortality, ABVD—doxorubicin, bleomycin, vinblastine and dacarbazine, AVD—doxorubicin, bleomycin and dacarbazine, IFRT—involved-field radiotherapy, PFS—progression-free survival, OS—overall survival, FFS—failure-free survival, NR—not reported. Footnotes: * not showing HL related outcomes which were superior; ** comparison with younger population; *** AE rate grade III or IV; **** data separated for arms in HD13 and HD 10 trial, IFRT used, *p* value not reported for outcomes with comparison to ABVD regimen in HD13, see text for explanation; ***** data shown for ABVD regimen.

**Table 3 biomedicines-10-02917-t003:** Alternative Chemotherapy Regimens with Curative Intent in Elderly Patients with Hodgkin Lymphoma.

Study	*n*	Disease Stage	Regimen	CR	Outcomes Rate	AE N, (Rate) *	CVD AE * N (Rate)	Respiratory AE N (Rate) *	Infection AE N (Rate) *	TRM Due to AEs	Discontinuation Rate/Deviation
Ballova et al. [56] **	42	advanced	BEACOPP baseline	76%	5-year FFTF= 46% 5-year OS = 50% ***	34 (87%) **	6 (15%)	3 (8%)	9 (23%)	21%	50%
Halbsguth et al. [57] ***	60	early advanced	BACOPP	85%	3-year PFS = 62% 3-year OS = 66%	52 (87%)	5(8%)	7(12%)	14(23%)	12%	30%
Kolstad et al. [60] ****	29	early advanced	CHOP21+ irradiation	93%	3-year PFS = 76% Median OS not reached	16 (51%)	2(7%)	1(3%)	9(31%)	7%	n.r.
Boll et al. [61] *****	59	early advanced	PVAG	78%	3-year PFS = 58.4% 3-year OS = 66.1%	43 (75.4%)	4 (7%)	4 (7%)	13 (22.8%)	1.7%	36%
Levis et al. [63] ******	48	early	VEBEMP	98%	5-year FFS = 79% 5-year OS = 94%	N/A	NR	NR	2 (5%)	0	3 (7%)
	57	advanced		58%	5-year FFS= 34% 5-year OS = 32%				8 (14%)	2 (3%)	15 (26%)

Abbreviations: N—number, CR—complete remission, AEs—adverse events, TRM—treatment-related mortality, BEACOPP—bleomycin, etoposide, doxorubicin, cyclophosphamide, vincristine, procarbazine and prednisone, FFTF—freedom from treatment failure, OS—overall survival, BACOPP—bleomycin, doxorubicin, cyclophosphamide, vincristine, procarbazine and prednisone, PFS—progression-free survival, PVAG—prednisone, vinblastine, doxorubicin and gemcitabine, VEBEMP—vinblastine, cyclophosphamide, procarbazine, etoposide, mitoxantrone and bleomycin, FFS—failure-free survival, N/A—not applicable, NR—not reported. Footnotes: * grade III or IV AES; ** RCT, comparator arm COPP-ABVD, data shown for BEACOPP baseline, total AEs reported as grade IV, *p* value not significant for CR, FFTF, OS and HL-related outcomes; *** high AE rate despite the extensive use of supportive care (G-CSF, ESA); **** retrospective design, small N, lack of supportive care number of cycles varied by disease stage and risk factors; ***** prospective study, treatment (number of cycles, radiotherapy) based on interim and definitive restaging, outcome data for whole group, HL, related outcomes superior, CVD grade V event after follow-up; ****** stratified by stage of cycles, IFRT, *p* value significant for CR, outcomes and AE.

**Table 4 biomedicines-10-02917-t004:** The Role of Brentuximab Vedotin in Elderly Patients Suffering from Hodgkin Lymphoma.

Study	N	Frailty Status	Disease Stage	Regimen	CR	Outcomes	AE Rate *	PN *	Infection AE Rate *	Discontinuation Rate **
Forero-Torres et al. [65]	26	Frail	Early Advanced	BV monotherapy ***	73%	Median PFS 10.5 mos Median OS not reached	NR	26%	NR	41%
Gibb et al. [66]	31	Frail	Advanced	BV monotherapy ***	26%	Median PFS 6. mos Median OS 19 mos	58%	10%	NR	29%
Yasenchak et al. [67]	26	Frail	Early Advanced	BV monotherapy ***	NR (ORR = 92%)	Median PFS 10 mos Median OS 77 mos	50%	NR	NR	42%
Yasenchak et al. [67] ****	22	Frail	Early Advanced	BV + DTIC	NR (ORR = 100%)	Median PFS 46 mos Median OS 64 mos	37%	NR	NR	42%
Schiano de Colella [69] *****	49	Frail	Advanced	BV + B	63%	2-year PFS = 54% 2-year OS = 83%	NR	Not recorded	CMV reactivation 17 events ******	16%
Evans et al. [71] *******	98	Fit	Advanced	ABVD	NR	2-year PFS = 68%	80%	NR	17%	BV discontinuation 20%
	83			BV-ABV		2-year PFS =73%	88%	NR	37%	
Boell et al. [72]	49	Fit	Advanced	BV-CAP	65%	1-year PFS = 74% 1-year OS = 92%	NR	Not recorded	27%	2 events
Evens et al. [73]	48	Fit Frail *******	Advanced	Sequential therapy	90%	2-year EFS = 80% 2-year OS = 93%	42%	4%	16%	35%

Abbreviations: N—number, CR—complete remission, AEs—adverse events, PN—peripheral neuropathy, BV—brentuximab vedotin, DTIC—dacarbazine, B—bendamustine, ABVD—doxorubicin, bleomycin, vinblastine and dacarbazine, BV+AVD—brentuximab vedotin, doxorubicin, vinblastine and dacarbazine, BV-CAP—brentuximab vedotin, cyclophosphamide, doxorubicin and prednisone, NR—not reported, PFS—progression-free survival, OS—overall survival, EFS—event-free survival, mos—months, CMV—cytomegalovirus. Footnotes: * grade III or V AEs; ** discontinuation rate due to toxicity, mainly PN; *** different study design; **** showing updated results, for more detail see text; ***** combination of BV+B early terminated in a study by Friedberg et al. [68]; ****** 4 grade V events; ******* analysis of ECHELON-1 RCT [70], stratification on ABVD or BV+AVD arms, for PN see text, *p* value not significant; ******* loss of IADL as a predictor of outcome, see text, Evens et al. [74].

## Data Availability

Not applicable.

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
