# Peer review of "Clinical Dilemmas in the Treatment of Elderly Patients Suffering from Hodgkin Lymphoma: A Review"

_biomedicines, 2022, doi:10.3390/biomedicines10112917_

Round 1

Reviewer 1 Report

In this paper the authors reviewed different clinical approaches in elderly/frail population with Hodgkin lymphoma. The review contains extended information about first line treatment, but unfortunaltelly lacks any novelty. Similar reviews have already been published several times (e.g. AM Evens, Blood 2019; Carter J, Curr. Treat. Options in Oncol. 2020) with very similar structure and practicaly identical pictures (figure 1). These reviews are cited in the text, however there is no reference regarding the figure, which, as already said does not bring any new information to the field. Recommendations after failing the first line treatment are missing.

Also, informations about new drugs are inconsistent and not up to date. E.g. pembrolizumab have been evaluated in KEYNOTE-204 trial, where 18% of patients were ≥65.

Author Response

Response to reviewer 1.

Reviewer 1. makes a point that similar articles were published by Evens et al.  Blood, 2019 and Carter et al., Cur. Treat. Options in Oncol. 2020 and this review article does add novelties to the field.

This review article has been prepared for special issue “Aging and Hematological Neoplasms” of journal Biomedicines, and, we believe that the topic of the article is appropriate. We were not able to describe novelties of the field lies due the fact that clinical trials in this population are scarce due to rigorous inclusion criteria excluding most of the “real-world” population. Furthermore, by writing the review article we have tried underline this problem and promote elderly Hodgkin lymphoma as unmet need in hematology to wider readership i.e., this is not a review per review’s sake often seen in science. Furthermore, we analyzed every modality of first-line treatment underlying PROs and CONs of modality used and, thus, have given a reader opportunity the critical overview of the field.

Reviewer 1. comments that figure 1. has no reference citation.

Figure 1. is original work made by Aron Belso, who is acknowledged in the appropriate area. All modalities shown in figure 1. are properly cited in the conclusion. Furthermore, the figure 1. is not reused from a particular paper and, therefore, no citation is needed.

Reviewer 1. comments that the references are not up to date suggesting the use of KEYNOTE-204 trial and that the treatment in relapsed-refractory setting is missing. The academic editor also suggested the comments on novel drugs in relapsed-refractory setting.

We have expanded the article with modalities used in relapsed-refractory setting ranging from potential role of high dose chemotherapy to conventional salvage treatment. We have included KEYNOTE 204 trial. However, there is a paucity of data in clinical trials in this setting. Although the elderly subjects are included in these trials, no formal analysis based on age are performed. For example, CheckMate 205 trial included 15 patients older than 60 years of age, no data on the efficacy of nivolumab in can be assessed. It is also true for AETHERA trial, although one may speculate that brentuximab vedotin consolidation in high risk fit patients.

Technical comments:

The final revision is submitted in landscape mode due to the fact that all tables are included which will be formatted in later stage.

Reviewer 2 Report

The authors undertook a comprehensive literature review on the treatment of elderly Hodgkin lymphoma patients, where prospective studies are scant and no clear guidelines exist. The abstract nicely summarises the paper. In the introduction a reference is missing for the death rate of patients older than 65.

In section 2, the title is missing (or is misplaced) for Table 1, that describes the geriatric assessment of elderly patients. This is an important table and a lot of information is described as footnotes, which makes reading difficult. The most relevant comments should be moved to the text. Some of them deserve discussion/are not clear (eg. **GA not predictive in HL in a recently published study).

Section 3 discusses the role of ABVD for the treatment of elderly patients. It should clearly separate the discussion concerning early/intermediate and advanced stages and identify the conclusions/recommendations. 

Subsection 7.5, dealing with the role of Brentuximab vedotin in first line treatment of elderly patients, should discuss the most relevant data contained in supplementary table 3.

Data on the results of different chemotherapy regimens and brentuximab monotherapy/combinations is presented as supplementary material. It might be of benefit for the reader that the main studies, that form the basis for the final recommendations in figure 1, would be presented in a table in the main text.

As minor points, there are some spelling mistakes in the footnotes of the supplementary tables. Those are sometimes difficult to follow and can be more detailed. Occasional references are not complete (eg. ref. 71)

Author Response

Response to reviewer 2.

The reviewer comments that a reference to HL patient death rate is missing in the introduction part.

The death rate of HL has been taken from S.E.E.R database and according reference has been assigned.

The reviewer comments that a name for a table 1. is missing. Furthermore, the reviewer comments that a table lacks clarity and that certain studies should be described in the main text, especially geriatric assessment in HL and the lack of association.

Appropriate name” The Geriatric Approach to Elderly Patients with Malignant Lymphoma” has been assigned. For further clarity, we have described studies by Eyre et al., 2021, Lee et al. 2021 and Yamasaki et al., 2022. The most emphasis was given on the work Orellana-Noia et al., 2021 due to the lack of GA and HL outcome and critical analysis of the pitfalls of the given study.

The reviewer suggest that section 3 should be divide into two sections “Early/intermediate disease” and “Advanced disease. Furthermore, it is suggested that treatment suggestion is given.

The section 3. is divided accordingly, while treatment recommendations are provided in section 5 “ABVD and Leading Guidelines”.

The reviewer recommends that the most relevant data on brentuximab vedotin is explained in subsection 7.5.

We have critically analyzed the data in subsection 7.5

The reviewer suggest that the supplementary tables are incorporated in the text.

All tables are incorporated in full text without supplementary tables left

The reviewer recognized the minor spelling and references mistakes.

The spelling mistakes have been corrected while references have been re-checked.

Technical comments:

The final revision is submitted in landscape mode due to the fact that all tables are included which will be formatted in later stage.

Round 2

Reviewer 1 Report

The changes improved the article appropriately. No other issues